# Does Self-Myofascial Release Cause a Remote Hamstring Stretching Effect Based on Myofascial Chains? A Randomized Controlled Trial

**DOI:** 10.3390/ijerph182312356

**Published:** 2021-11-24

**Authors:** Paul Fauris, Carlos López-de-Celis, Max Canet-Vintró, Juan Carlos Martin, Luis Llurda-Almuzara, Jacobo Rodríguez-Sanz, Noé Labata-Lezaun, Mathias Simon, Albert Pérez-Bellmunt

**Affiliations:** 1Faculty of Medicine and Health Sciences, Universitat Internacional de Catalunya, 08017 Sant Cugat del Vallès, Spain; pfauris@uic.es (P.F.); carlesldc@uic.es (C.L.-d.-C.); maxcanet44@uic.es (M.C.-V.); jcmartin@uic.es (J.C.M.); lllurda@uic.es (L.L.-A.); jrodriguezs@uic.es (J.R.-S.); nlabata@uic.es (N.L.-L.); msimon@uic.es (M.S.); 2ACTIUM Functional Anatomy Group, Universitat Internacional de Catalunya, 08195 Sant Cugat del Vallès, Spain; 3Institut Universitari per a la Recerca a I’Atenció Primària de Salut Jordi Gol i Gurina (IDIAPJGol), 08007 Barcelona, Spain

**Keywords:** fascia, flexibility, hamstring muscles, self-myofascial release, meridians

## Abstract

Background: The hamstring muscles are described as forming part of myofascial chains or meridians, and the superficial back line (SBL) is one such chain. Good hamstring flexibility is fundamental to sporting performance and is associated with prevention of injuries of these muscles. The aim of this study was to measure the effect of self-myofascial release (SMR) on hamstring flexibility and determine which segment of the SBL resulted in the greatest increase in flexibility. Methods: 94 volunteers were randomly assigned to a control group or to one of the five intervention groups. In the intervention groups, SMR was applied to one of the five segments of the SBL (plantar fascia, posterior part of the sural fascia, posterior part of the crural fascia, lumbar fascia or epicranial aponeurosis) for 10 min. The analyzed variables were hamstring flexibility at 30 s, 2, 5, and 10 min, and dorsiflexion range of motion before and after the intervention. Results: Hamstring flexibility and ankle dorsiflexion improved when SMR was performed on any of the SBL segments. The segments with the greatest effect were the posterior part of the sural fascia when the intervention was brief (30 s to 2 min) or the posterior part of the crural fascia when the intervention was longer (5 or 10 min). In general, 50% of the flexibility gain was obtained during the first 2 min of SMR. Conclusions: The SBL may be considered a functional structure, and SMR to any of the segments can improve hamstring flexibility and ankle dorsiflexion.

## 1. Introduction

Classic anatomy textbooks describe the hamstring muscles from a traditional mechanistic view as being isolated from the adjacent structures [1,2,3] However, recent research has changed this perspective, proposing a connective tissue link between the active components of the movement system to form an extensive network of myofascial chains, or meridians [4]. Of these meridians, there is most anatomical evidence for the superficial back line (SBL) [4,5]. This myofascial chain, described by Myers, connects the hamstring muscles with the gastrocnemius muscles and the plantar fascia caudally, and with the thoracolumbar fasciae, the erector spinae muscle and the epicranial aponeurosis cranially [6]. Functionally, anatomical studies have demonstrated the role of these fascia in the transmission of biomechanical force between the different regions of the body connected by the chains [7,8,9,10,11].

The hamstring and gastrocnemius muscles have a fundamental role in sports performance; hamstring injuries are common and associated with a high rate of recurrence [12,13,14]. As a result, the etiology of a hamstring injury has been extensively researched, and several risk factors have been proposed. Some of these studies have reported that reduced hamstring flexibility is one of many factors that can predispose to hamstring muscle strains [14,15,16,17,18,19]. Previous investigations have pointed out that a lower range of motion (ROM) or flexibility [20,21] and increased muscle stiffness [22,23] are risk factors for sports injuries [24]. Research has shown that increased muscle stiffness associated with antagonist muscle contractions can inhibit joint movement and result in higher energetic/metabolic costs [25]. Therefore, it could be essential to improve ROM and decrease muscle stiffness in sports and rehabilitation settings.

With the objectives to improve the ROM [26,27,28] and flexibility [29,30] and reduce the stiffness and pain [31] in the rehabilitation and sport field appear the self-myofascial release (SMR). Self-myofascial release (SMR) uses a foam roller involves the pressure of a person’s own body weight to release tension in muscles, tendons, fascia, and/or soft tissues.

Although there is anatomical and functional evidence of some of these chains, to our knowledge no study has demonstrated a remote effect on hamstring or gastrocnemius flexibility from the use of a foam roller at distinct segments of the superficial back line (SBL). The present study aimed to address this research deficit, by measuring hamstring and gastrocnemius flexibility improvement when self-myofascial release was applied to the various segments of the SBL and determining—if such a remote effect were present—the segment and duration of treatment that resulted in the greatest increase in flexibility. If this functional unit exists, this study could make it possible to propose new remote therapeutic strategies.

## 2. Materials and Methods

### 2.1. Participants

The participants recruited for the present study were healthy subjects. Of the 105 volunteers who were initially screened, 94 were enrolled. All details of the enrolment process appear in Figure 1.

Approval was obtained from the local ethics committee (CBAS1805, 1 April 2018). and the study was conducted in accordance with the declaration of Helsinki. The trial was registered at clinicaltrials.gov. All participants gave written informed consent prior to testing.

Volunteers were asked to abstain from alcohol for 24 h priors to participation. Exclusion criteria were age below 18 years, surgical procedures in the previous 6 months, overweight or obese, previous hamstring injury or musculoskeletal problems in the previous 12 months, hypermobility, severe neurological or cardiovascular disease, pregnancy, and existing muscle soreness.

### 2.2. Study Design

The study was a single-blind randomized control trial (clinicaltrials.gov, accessed on 11 May 2018, NCT03521544). Interventions and measurements were carried out by the same 2 blindedtherapists, one controlling the intervention and the other taking measurements and recording data. To minimize bias, participants could not see and were not informed of any result obtained during measurement. A local committee approved this study (CBAS1805).

### 2.3. Variables and Measurements

Hamstring muscle flexibility was evaluated with a modified sit-and-reach test (MSR) before, during (at 30 s, 2 and 5 min) and at the end of the intervention (at 10 min).

This test was selected based on its wide use by clinicians to evaluate hamstring flexibility and its convenient correlation coefficient [32,33,34]. It was carried out following the recommendations in previous published research [33,35]. The participants sat on the floor with the lower limbs stretched out and together, the back and hips supported against the wall (90° hip flexion), and the soles of the feet placed against the edge of the box. Participants then extended their arms forward with the same hand on top of the other facing down, keeping their back against the wall. They then reached forward, sliding their hands along the measuring scale as far as possible without bending the knees [36].

Dorsiflexion range of motion of the ankle was analyzed with the dorsiflexion lunge test (DF-lunge), out before and after the intervention following the recommendations of previous investigations [29,37]. The test is performed by placing the foot perpendicular to a wall and lunging the knee towards the wall. The foot is sequentially moved farther away from the wall until the maximum range of dorsiflexion is achieved. As the heel should not lift off the floor, a band was placed under the heel and tension applied by the therapist, so that if the heel raised off the floor the band would snap loose. The distance from the tiptoe to the wall was measured. This test has shown a good intra- and inter-rater reliability in healthy subjects [38].

Subjective perception of physical effort exerted during the intervention. The modified Borg scale is a standardized visual analog scale that assesses subjective perception of breathing difficulty (dyspnea) or physical effort exerted [39]. This consisted of a vertical scale labelled from 0 to 10, with corresponding verbal expressions of progressively increasing perceived sensations of intensity [40].

### 2.4. Interventions

After enrolment, the participants signed an informed consent form and immediately after this, baseline measurements were taken. At this point each participant was randomly assigned to the control group or one of the five experimental groups using a random number table.

#### 2.4.1. Control Group

The control group did not receive any intervention; participants lay on a treatment table for the same amount of time as the intervention lasted in the experimental group (10 min). The data on hamstring muscle flexibility were collected at the start and at 30 s, 2, 5 and 10 min. Ankle dorsiflexion was measured at the start and at the end of the intervention.

#### 2.4.2. Intervention Groups

Five experimental groups were created, based on the SBL segment that was to receive SMR (plantar fascia, posterior part of the sural fascia, posterior part of the crural fascia, lumbar fascia and epicranial aponeurosis) (Figure 2). A physical therapist with more than 10 years of experience in manual therapy and soft tissue techniques gave a theoretical and practical explanation to the participants before the intervention. This person ensured during the intervention that the participants performed the intervention well. The intervention followed the recommendations of previous investigations and the applied pressure was the maximum tolerated by the patients (pushing into discomfort) with no pain [41]. Data collection on ankle flexibility and dorsiflexion (ROM) were measured the same way as the control group. Subjective perception of effort was analyzed at the end of the intervention in each group.

#### 2.4.3. Statistical Analysis

Variables were described as mean and standard deviation for quantitative variables and frequency and percentage for qualitative variables. For the mean flexibility gain, a 95% confidence interval was calculated. For k–group comparison of independent samples, ANOVA was performed for quantitative variables and Chi-square test for quantitative variables.

## 3. Results

The groups did not differ regarding age, gender or baseline measurements of ankle ROM or hamstring muscle flexibility (*p* > 0.05, Table 1).

At the end of the intervention, it was observed that application of SMR to the superficial back line resulted in a statistically significant improvement in hamstring muscle flexibility (Table 2).

The intervention groups had an improvement of 5.22 cm, while the control group had an improvement of 2.93 cm. Ankle ROM also improved on the DF-lunge test after the intervention (Table 2 and Figure 3).

When comparing the results from the different SBL segments, to determine the optimal location for SMR for hamstring flexibility, we found two different situations depending on the application time. For short interventions (2 min or less of SMR), the best results were obtained with SMR to the plantar fascia and the posterior part of the sural fascia (Table 2 and Figure 3). For longer interventions (5 or 10 min of SMR), more flexibility was gained with SMR to the posterior part of the sural fascia. Good results were also obtained with SMR to the thoracolumbar fascia and the posterior part of the crural fascia (Table 2 and Figure 4).

Regarding intervention duration, overall, 70.5% of hamstring flexibility gain was obtained with 5 min of SMR application, but after 30 s 23.5% of the total improvement could already be seen (Table 3).

When analyzing intervention duration according to the treated zone, we found that when SMR was performed on the sole of the foot, 70% of the gain was obtained at 2 min, while all the other zones had around a 50% gain after 2 min (Table 3 and Figure 4). No association was observed between perceived effort and flexibility gain (Table 2).

## 4. Discussion

The present study found that performing SMR on any segment of the SBL resulted in a statistically significant increase in hamstring flexibility and ankle dorsiflexion. These results reinforce the concept of the chain as an entity, not just from an anatomic perspective, as has been described previously [4,5], but also as a functional structure as reported in recent studies [42].

The segment found to affect hamstring flexibility most was the plantar fascia. The effects were greater especially when the foam roller was applied for relatively short times. This implication is not new and the results are similar to those observed by Grieve et al., when self-massage of the plantar fascia with a ball was performed for 5 min [43], or to the results described by Do et al. with self-massage for 5 min [44]. Joshi et al. obtained comparable results to ours when treatment was applied by a therapist over various sessions; they also found better results when the SMR was self-applied [42]. Even so, the results from this study are slightly lower than those described by Patel et al. [45].

Our results show that when the foam roller was applied for longer periods of time (from 5 to 10 min), the segments of the thoracolumbar fasciae and erector muscles, and the posterior part of the sural fascia had more impact on hamstring flexibility. Previous studies observed an improvement in dorsiflexion but did not assess hamstring flexibility [46]. Other investigations also found increased ankle dorsiflexion following self-massage of the posterior part of the lower limb [47], but did not establish which fascial segment had the greatest effect.

The effects of SMR to each of these segments on hamstring flexibility may be due to the continuity, overlapping and compartmentalization of the muscular tissue by the fascia.

A possible explanation for the better results when SMR was performed on the plantar fascia is the significant fascial continuity, via the Achilles tendon, between the plantar fascia and the posterior part of the sural fascia [10]. In the same way, SMR application to the thoracolumbar fascia and erector muscles can be explained by the anatomical connection between the thoracolumbar fascia and the hamstring muscles via the sacrotuberous ligament [9,48]. Other studies have found significant improvements by combining the foam roll with passive movements [49]. However, it appears that the use of other therapies such as focal vibration may generate superior improvements [50]. Despite these results, the selling price of focal vibration is much higher than that of foam roll. For this reason, it could be interesting to use therapies such as focal vibration within a therapeutic treatment with the physical therapist and the use of the foam roll as a self-treatment for the patient.

Regarding the results obtained, there is a fascial structural relationship in the entire posterior chain; however, there is also a neurophysiological explanation. This possible explanation for the results obtained could be the one proposed by Bialosky JE et al. [51]. This author states that different manual techniques can generate viscoelastic and hypoalgesic effects through a model based on the fact that a mechanical stimulus initiates a chain of spinal, peripheral, and/or supraspinal neurophysiological events that would produce these changes at a distance even in distant areas such as the suboccipital region [52].

In summary, the present study demonstrates that the SBL can be considered a functional structure, since SMR application to any of its component segments improved hamstring flexibility and ankle dorsiflexion, the segments with the greatest effects being the plantar fascia (for short treatment duration) and the thoracolumbar fascia and erector muscles (for longer treatment duration). The results suggest that this technique could be effective both in rehabilitation and sports to enhance hamstring flexibility remotely.

The main limitation of this study is that short- and medium-term follow-ups have not been carried out, so we do not know how long the effects of the techniques used are maintained. Another limitation of this study is that more precise measurement tools such as myotonometry or tensiomyography could have been used. For future studies, the combination of functional measurements with these tools and short- and medium-term follow-up is recommended.

## 5. Conclusions

The present study demonstrates that the SBL can be considered a functional structure, as the application of SMR on any of its component segments improved hamstring flexibility and ankle dorsiflexion. The use of this treatment could be effective in both rehabilitation and sport to improve hamstring flexibility at a distance.

## Figures and Tables

**Figure 1 ijerph-18-12356-f001:**
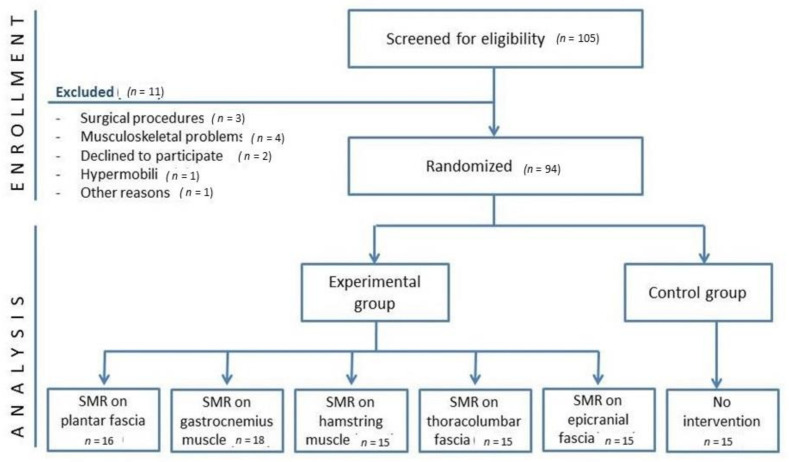
Flow chart of the trial.

**Figure 2 ijerph-18-12356-f002:**
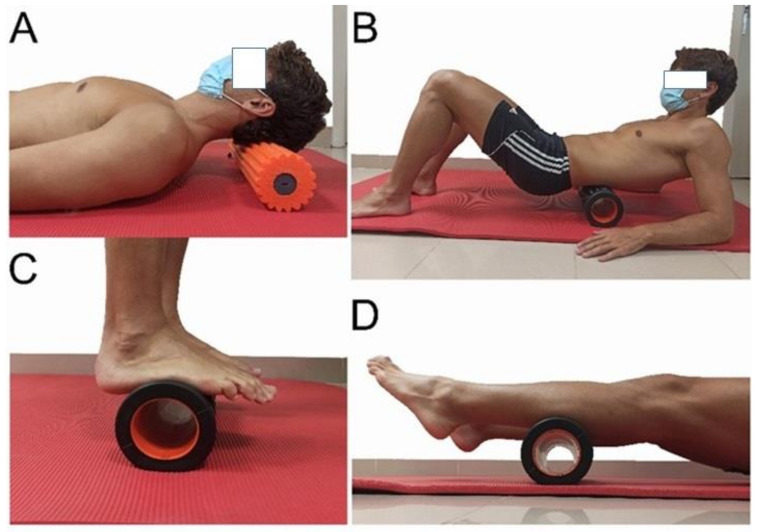
Interventions. (**A**) intervention to the epicranial aponeurosis, (**B**) thoracolumbar facia and erector muscles intervention, (**C**) plantar fascia and (**D**) posterior part of the sural fascia.

**Figure 3 ijerph-18-12356-f003:**
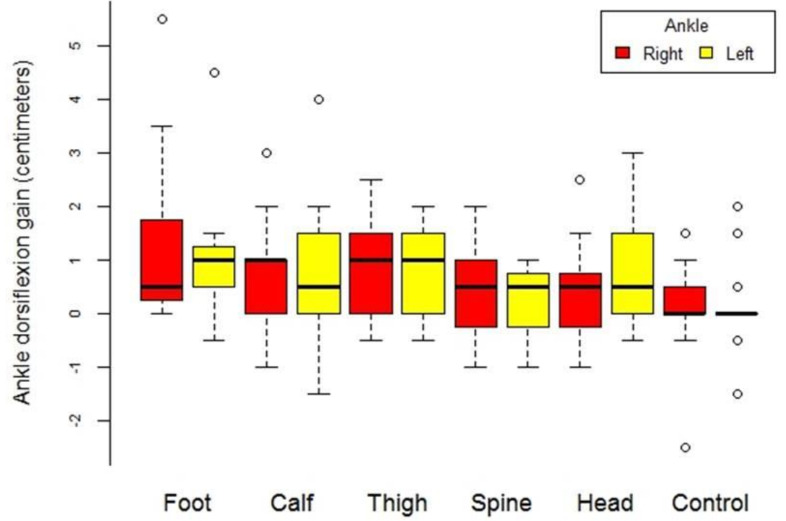
Box diagram of the gain of increase in ankle flexion-extension of the ankle by location of treatment location and by extremity (right and left).

**Figure 4 ijerph-18-12356-f004:**
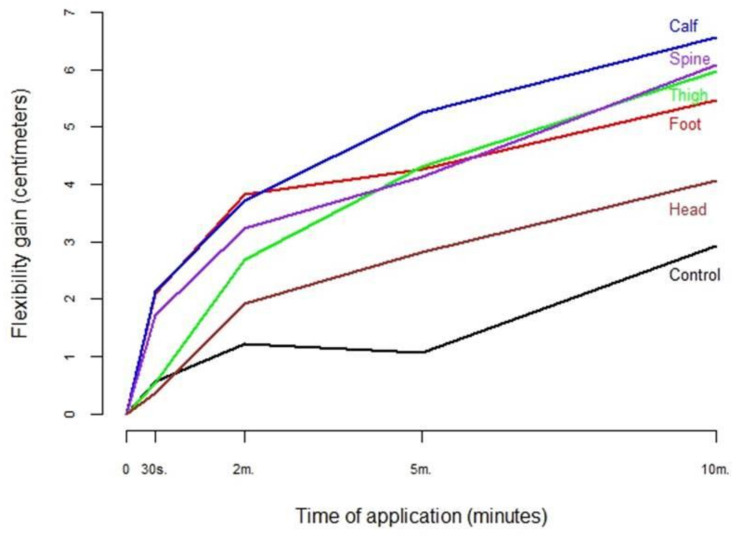
Hamstring flexibility gain (cm) according to group.

**Table 1 ijerph-18-12356-t001:** Baseline characteristics of the participants grouped and compared by their intervention. Mean (standard deviation) for the quantitative variables; Frequency (percentage) for the qualitative variables; ANOVA test for the quantitative variables; Chi-square for the qualitative variables.

Sample Size		Age	Gender		Hamstring Flex.	Dorsiflexion	
			Women	Men		Right	Left
Total	94	24.5 (5.6)	42 (43.6%)	52 (56.4%)	25.1 (10.9)	10.3 (3.5)	10.3 (3.6)
Intervention							
Control group	15	26.1 (6.4)	46.70%	53.30%	28.0 (12.0)	10.5 (2.5)	10.6 (3.2)
Plantar fascia	16	25.0 (7.5)	60.00%	40.00%	24.8 (13.2)	9.3 (4.2)	9.2 (4.3)
Posterior part of the sural fascia	18	24.8 (5.3)	18.80%	81.30%	21.9 (11.0)	11.3 (3.5)	11.7 (3.9)
Posterior part of the crural fascia	15	23.8 (5.4)	61.10%	38.90%	27.9 (11.0)	9.0 (2.8)	9.1 (1.9)
Thoracolumbar fasciae and erector muscles	15	23.2 (4.2)	26.70%	73.30%	24.1 (10.6)	11.1 (3.6)	11.2 (3.6)
Epicranial aponeurosis	15	23.5 (6.4)	46.70%	53.30%	23.4 (7.4)	11.0 (3.7)	10.5 (4.4)
*p*-value		0.521	0.082		0.545	0.232	0.245

**Table 2 ijerph-18-12356-t002:** Increase in hamstring flexibility (cm) for each time interval and increase in dorsiflexion of ankle for each group (mean (95% CI)).

	Sit-and-Reach Test. Increase in Hamstring Flexibility				Lunge Test. Increase in Ankle Dorsiflexion		Gain vs. Perception of Physical Effort
Group	0–30 s	0–2 min	0–5 min	0–10 min	Right	Left	Effort, Mean (SD)
Control group	0.56 (−0.31; 1.45)	**1.23 (0.29; 2.17)**	1.07 (−0.21; 2.34)	**2.93 (0.28;5.58)**	0.13 (−0.36; 0.63)	0.14 (−0.30; 0.57)	
Intervention groups	**1.23 (0.82; 1.65)**	**2.78 (2.24; 3.31)**	**3.68 (2.98; 4.38)**	**5.22 (4.39; 6.05)**	**0.64 (0.42; 0.86)**	**0.66 (0.45; 0.86)**	
Plantar fascia	**2.10 (0.73; 3.46)**	**3.83 (2.26; 5.40)**	**4.27 (2.45; 6.08)**	**5.47 (3.38; 7.55)**	**1.23 (0.38; 2.08)**	**1.03 (0.41; 1.65)**	**1.4 (1.7)**
Posterior part of the sural fascia	**2.15 (1.12; 3.18)**	**3.71 (2.59; 4.84)**	**5.25 (3.81; 6.69)**	**6.56 (4.51; 8.61)**	**0.75 (0.24; 1.26)**	**0.78 (0.11; 1.46)**	**3.1 (0.7)**
Posterior part of the crural fascia	**0.55 (−0.003; 1.11)**	**2.69 (1.43; 3.95)**	**4.31 (2.72; 5.89)**	**5.97 (4.22; 7.72)**	**0.86 (0.43; 1.28)**	**0.94 (0.54; 1.35)**	**2.1 (0.7)**
Thoracolumbar fasciae and erector muscles	**1.73 (0.83; 2.63)**	**3.23 (1.83; 4.63)**	**4.13 (1.90; 6.36)**	**6.07 (4.22; 7.91)**	0.43 (−0.13; 0.99)	0.23 (−0.12; 0.59)	**1.2 (0.8)**
Epicranial aponeurosis	**0.37 (−1.03; 1.76)**	**1.93 (0.25; 3.62)**	**2.83 (0.79; 4.86)**	**4.07 (1.62; 5.91)**	0.37 (−0.12; 0.86)	0.77 (0.16; 1.37)	**0.8 (1.0)**

Bold: mean statistically significantly different from baseline.

**Table 3 ijerph-18-12356-t003:** Percentage of total flexibility increase achieved at each stop-time.

	30 s	2 min	5 min	10 min
Intervention groups overall	23.50	53.2	70.5	100
Plantar fascia	38.4	70	77.8	100
Posterior part of the sural fascia	32.7	56.6	80	100
Posterior part of the crural fascia	9.2	45	72	100
Thoracolumbar fasciae and erector muscles.	28.0	52.8	67.7	100
Epicranial aponeurosis	9.1	47.6	69.8	100

## Data Availability

The data presented in this study are available on request from the corresponding author.

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
