# Peer review of "Does Self-Myofascial Release Cause a Remote Hamstring Stretching Effect Based on Myofascial Chains? A Randomized Controlled Trial"

_ijerph, 2021, doi:10.3390/ijerph182312356_

Round 1
Reviewer 1 Report
Despite some advantages of the study, I am concerned that this manuscript should not be published in its current form. Major changes are needed. I present suggestions for changes for each chapter below.
In the study, the authors used the following tests: modified sit-and-reach and dorsiflexion range of motion of the ankle. These tests cover different anatomical regions and test different muscle groups. The research goal itself indicates the assessment of hamstrings muscles in the concept of myofascial chains, so I do not understand why the authors used the dorsiflexion range of motion of the ankle test. The use of functional tests to study completely different muscle groups (despite the fact that these muscles are part of the superficial back line) forces the title of the manuscript and the aim of the study to be changed.
Were participants trained to perform SMR? If so, who gave the instructions? Was it a specialist in this field? What was his professional experience and was he trained in soft tissue techniques? This information should be added to the manuscript in the material and methods section.
Why did the researchers choose to use only functional muscle tests using a centimeter measure to assess flexibility? In today's research on the functional and biophysical aspects of selected muscles, more advanced research methods (e.g. electrogoniometers, motion capture systems, myotonometry) are used, which allow to provide more detailed data from the examined muscles.
In the material and methods section (lines 89-93), the authors described the modified sit-and-reach test used: "… The participants sat on the floor with the lower limbs stretched out and together, the back and hips supported against the wall (90 ° hip flexion), and the soles of the feet placed against the edge of the box. Participants then extended their arms forward with the same hand on top of the other facing down, keeping their back against the wall… ”. In the presented Figure 2. "Evaluation of hamstring flexibility by modified sit-and-reach test. Initial position (A) and final position (B) of the test "the starting and ending positions look completely different. There is no protection of the lumbar spine against the wall, which means that the participant flexed the spine during the test, and as a result, the test did not only concern hamstrings muscles. Please explain these inconsistencies.
I believe that follow-up measurement should be included in this type of research. It is important because it provides valuable data that allow to assess whether the obtained effects are only short-term. Such a measurement may, for example, be performed on the day following the intervention. This type of data allows to determine whether the obtained effects are sustained, which allows to determine the required frequency of a given therapy.
The discussion is definitely insufficient. There is no explanation of the mechanisms responsible for the effect of the applied SMR therapy. The authors focused on aspects related to the myofascial chains themselves (this is also important), but omitted the autotherapy itself. The comparison of the obtained results with other studys on similar topics is sparse and general. I also propose to create an additional paragraph in which a comparison of the described SMR therapy with other soft tissue therapies in the context of myofascial chains would be included. At the end of the discussion, I propose to create a separate paragraph in which the authors would include information related to research limitations and suggestions for future studies. This is quite an important paragraph pointing the way for future research in this scientific field.
Conclusions section must be corrected. The authors described this important chapter in the form of an overview of the results. Conclusions must contain the most important information explaining these results.
Reviewer 2 Report
Please fill in the Ethics Review Committee's approval number.
1. Although the paper is generally well written, it is apparent in some areas that the authors’ first language is not English. They should seek assistance from this perspective as part of the revision process. Some of the figure legends could provide a little more explanation. Please correct typos, ambiguous expressions, and unclear expressions, and check whether the overall grammar is clearly met. And please double-check that the format of this journal is correct.
2. It seems that a scientific basis for the methodological part that distinguishes MFR.Generally, this part of concept is often well-known in clinical practice, but I think that scientific content and the basis for it should be clear at least in this kind of academic papers. The distinction among plantar fascia, GCM, hamstring, thoracolumbar epicranial fascia and the rationale for whether the method is really a molecularly feasible concept is needed.
3. The concepts of self-MFR in the introduction and discussion part are lacking both quantitatively and qualitatively. There are many related studies, so please check and supplement them.
Author Response
Please see the attchment.

Round 2
Reviewer 1 Report
The corrections made significantly improved the scientific quality of the manuscript.